# Assessment of the Plans to Optimize Antimicrobial Use in the Pediatric Population in Catalan Hospitals: The VINCat Pediatric PROA SHARP Survey

**DOI:** 10.3390/antibiotics12020250

**Published:** 2023-01-26

**Authors:** Borja Guarch-Ibáñez, Aurora Fernández-Polo, Sergi Hernández, Eneritz Velasco-Arnaiz, Montse Giménez, Pere Sala-Castellvi, Valentí Pineda, Susana Melendo

**Affiliations:** 1Pediatric Infectious Diseases Unit ICS-IAS, Department of Pediatrics, Hospital Universitari de Girona Dr. Josep Trueta, Universitat de Girona, 17007 Girona, Spain; 2Department of Pharmacy, Hospital Universitari Vall d’Hebron, Vall d’Hebron Research Institute, Universidad Autónoma de Barcelona, 08035 Barcelona, Spain; 3VINCat Program Surveillance of Healthcare Related Infections in Catalonia, 08907 Barcelona, Spain; 4Antimicrobial Stewardship Program (PROA-SJD), Pediatric Infectious Diseases Unit, Department of Pediatrics, Hospital Sant Joan de Déu, 08950 Barcelona, Spain; 5Department of Microbiology, Laboratori Clínic de la Metropolitana Nord, Hospital Germans Trias i Pujol, 08916 Badalona, Spain; 6Department of Paediatrics, Hospital Universitari General de Catalunya, 08915 Sant Cugat del Vallés, Spain; 7Pediatric Infectious Diseases Unit, Department of Pediatrics, Hospital Universitari Parc Taulí, Universidad Autónoma de Barcelona, 08208 Barcelona, Spain; 8Pediatric Infectious Diseases and Immunodeficiencies Unit, Hospital Infantil Vall d’Hebron, Vall d’Hebron Research Institute, 08035 Barcelona, Spain

**Keywords:** antibiotics, antimicrobial stewardship, quality improvement, pediatrics

## Abstract

In Spain, many programs have been introduced in recent years to optimize antimicrobial stewardship in pediatric care (known as pediatric PROA). However, information on the current situation of these programs is scarce. The present study assesses current antimicrobial use in pediatric care in the hospitals of Catalonia affiliated with the VINCat pediatric PROA group. Between December 2020 and January 2021, an electronic survey related to the design and use of PROA was administered to members of PROA teams in our hospital network. The survey was conducted at 26 hospitals. Twelve percent of the hospitals had pediatric PROA in operation, 42% were included in adult PROA, and 46% carried out pediatric PROA activities but not as part of an established program. At 81%, the pediatric PROA team included a pediatrician, in 58% a pharmacist, and in 54% a microbiologist. The main activities were monitoring the use of antimicrobials and bacterial resistance. Twenty-seven percent measured indicators regularly. The VINCat Pediatric PROA group’s hospitals have implemented measures for optimizing antimicrobial stewardship, but few have a pediatric PROA program in place. Specific measures and indicators must be defined, and the resources available should be increased. The development of pediatric PROA should be monitored in the coming years.

## 1. Introduction

The emergence of antimicrobial resistance represents a serious problem for public health systems all over the world. In fact, in 2019, the WHO listed antimicrobial resistance among the ten main threats facing global health [1]. Inappropriate use is one of the main causes of antimicrobial resistance [2,3]. One of the strategies to combat the problem has been the establishment of programs to optimize antimicrobial stewardship [4,5]. In Spain, programs of this kind were initiated in the consensus document by Rodríguez-Baño et al. published in 2012, which outlined the conditions required for their implementation and made recommendations for their further development [6]. More recently, in 2022, the Spanish Society of Paediatric Infectious Diseases published their position statement about the implementation of paediatric antimicrobial stewardship programs, providing tools to facilitate their application in hospitals throughout the regional health care systems in Spain [7]. In this country, these programs are known as PROA, an acronym of the title in Spanish, *PRogramas de Optimización de Antimicrobianos.*

In pediatric populations, infectious diseases are among the main causes of hospitalization and consultation, and antibiotics are the main therapeutic option in primary and hospital care [2,8]. The specific features of infectious diseases in pediatric patients, and the differences in the efficacy and safety of antimicrobials in this population with respect to adults, have contributed to the significant increase in PROA designed for children in recent years [2,9,10,11,12,13,14,15].

Since its creation in 2006, the Catalan Health Care-Related Infections Surveillance Program (VINCat) has assessed a series of indicators related to the use of antimicrobials [16,17,18]. In 2012, VINCat launched a program for optimizing antimicrobial use in the adult population (the VINCat hospital PROA) which it continues to coordinate today. Later, in October 2019, the VINCat pediatric PROA program was created, involving the vast majority of hospitals in Catalonia that care for pediatric patients as well as their corresponding primary care areas. The “core group” of coordinators comprises eight specialists in microbiology, pharmacy, and pediatrics, representative of the various levels of primary care and hospital care in Catalonia.

Several studies have evaluated the implementation and development of pediatric antimicrobial stewardship programs, above all in the US [19,20,21,22]. In Spain, information on the current state of pediatric PROA is scarce, both at national and at regional level [10,14,15,19]. The objective of this study is to assess the current status of pediatric PROA in the network of hospitals belonging to the VINCat pediatric PROA group, outlining the resources available for their implementation and development and describing the various activities carried out. The idea is that the results of our study should serve as a starting point for detecting possible areas in need of improvement, for standardizing indicators and for defining interventions able to address existing problems.

## 2. Results

The survey was sent to 52 hospitals that serve the pediatric population. Forty-three belonged to the Catalan public health system and nine were private hospitals.

Twenty-six responses were received (21 public and five private). These hospitals receive 78.9% of pediatric acute hospital admissions in Catalonia, according to data provided by the Catalan health service [23].

The survey was answered by a pediatrician at 19 of the 26 hospitals and by a pharmacist at seven. Eight hospitals (31%) were classed as level 1, ten (38%) level 2, and eight (31%) level 3. At 24 of the 26 hospitals, the pediatric service was located inside an adult hospital, while the remaining two were children’s hospitals. Fourteen hospitals (54%) reported that their pediatric service only saw patients up to 15 years of age, the maximum age for pediatric patients in primary care in Spain.

The median number of pediatric beds was 25 (IQR 10-249), including the general pediatric ward, pediatric medical and surgical specialties, intermediate care, and short-stay units. The median number of PICU and NICU beds was eight (IQR 3-24) and 12 (IQR 3-45) respectively. The median admission rate in pediatric medical services in 2019 was 973 admissions (IQR 276-9435) (see Figure 1 for more details on admissions).

### 2.1. Structure Indicators

Overall, 14 of the 26 hospitals surveyed (54%) had an antimicrobial stewardship program in place. Three (12%) had a stand-alone pediatric PROA program with institutional support; at 11 (42%), the pediatric program was integrated in the adult program, and in the remaining 12 (46%), there was no established pediatric PROA, but recommended antimicrobial stewardship actions were carried out. At 13 hospitals (50%), the interventions and results were reported to the institution’s infectious diseases commission or antibiotics commission, and at six (23%), to the adult PROA; at the other seven hospitals (27%), there was no record of the results being reported.

The structure indicators of each of the participating hospitals are summarized in Table 1. Eleven PROA teams (42%) comprised pediatricians, microbiologists, and pharmacists, while other specialists included were nurses (present in seven teams), neonatologists (in nine), pediatric intensivists, pediatric surgeons, and management representatives in three teams each, preventive medicine specialists in five, and data analysts in one.

Twenty-four out of 26 hospitals had computerized support for the prescription of antimicrobials.

### 2.2. Process Indicators and Antimicrobial Stewardship Activities

The stewardship activities carried out most frequently by hospital specialists were the supervision of the use of antimicrobials in pharmacy services (performed by 69%) and the monitoring of bacterial resistance in microbiology services (performed by 62%). The other activities undertaken are described in Table 2.

The actions carried out by PROA pharmacists were: monitoring of antimicrobial prescription (19/26), dose adjustment and optimization (20/26), and pharmacokinetic monitoring of antimicrobials (24/26). The anti-infectives most frequently monitored were vancomycin (22/24) and aminoglycosides (19/24), and to a lesser extent, voriconazole (8/24) and posaconazole (5/24). Other stewardship activities carried out at the various hospitals are described in Table 3.

As regards data analysis, seven hospitals (27%) routinely analyzed the indicators established for the pediatric antimicrobial stewardship program, including days of therapy (DOT) per 100 patient days. Thirteen (50%) recorded adverse reactions related to the use of antimicrobials, and 20 (77%) monitored microbial resistance in pediatric patients both in the hospital and in the primary care.

Details of the training activities related to the pediatric antimicrobial stewardship program at the hospitals are shown in Figure 2.

## 3. Discussion

The VINCat pediatric PROA Sharing Antimicrobial Reports for Pediatric Stewardship (SHARP) survey is the first description of the current state of pediatric antimicrobial stewardship programs in hospitals in Catalonia. Due to the high representativeness of the participating centers, this study should be a useful starting point for the assessment of pediatric PROA in Catalan hospitals both now and in the future, and for detecting areas in need of improvement.

The PROA should be considered as part of the institutional programs of the hospitals where they are performed [6,7]. The structure of the programs in place in the different center in this region varies widely, with the higher-level hospitals being the ones with the most developed programs. In any case, the design of the programs must meet certain minimum requirements with regard to their structure, objectives, and activities, and these will be influenced to some extent by the nature of the hospital or health area concerned, the organizational context, and the resources available.

Currently, the pediatric PROA in Catalonia tend to take the form of specific actions recommended in the stewardship programs or integrated within the adult programs. In more than a quarter of the cases, the interventions and results are not reported or are unknown. As regards the composition of the teams, PROA tend to have a larger presence of pediatricians than of pharmacists and microbiologists; interestingly, at 11 hospitals (42%), the PROA teams also comprise other health professionals, especially neonatologists.

The responses to the survey also reveal that the microbiological and imaging diagnostic services of our hospitals have their own computerized laboratories and perform most of the tests themselves. The main process indicators associated with the pediatric PROA are monitoring the use of antimicrobials, but only a small percentage of them regularly assess the pediatric program’s pre-established indicators.

The drugs in which pharmacokinetic monitoring is most frequently performed are vancomycin and aminoglycosides, due to their wide use in neonatal and pediatric intensive care units. At almost two thirds of the hospitals, educational audits are carried out with feedback in real time; this is a positive finding, since although their effects are not as immediate as those of restrictive interventions, they are more widely accepted by prescriptors [6]. In addition, the vast majority of hospitals carry out activities to review and update protocols, although the frequency of their performance varies widely from center to center. Only 14 (54%) have a protocol for surgical prophylaxis, which is one of the main sources of inappropriate antibiotic use [24]. Interestingly, 16 of them (62%) prescribe antimicrobials with a specific end date, which limits the administration of unnecessary doses.

Eleven of the hospitals (43%) do not provide any training in pediatric PROA, and only the most active and highest-level hospitals offer any advisory services. It is important to increase training in antimicrobial use at hospitals of all levels.

Pediatric antimicrobial stewardship programs in Catalonia have expanded in recent years but, as elsewhere in Europe, they lag behind similar programs in place in the US [19]. Studies in the US present data from hospitals with much higher numbers of pediatric beds and of annual admissions, and so our results can hardly be compared to theirs [20]. Nevertheless, our survey shares certain similarities with the initial situation in the US in 2011 when the SHARP survey was introduced there [20]. Specifically, pediatricians devote more time to the program than pharmacists (and in our case, microbiologists as well), and in the early stages, data analysts were not heavily involved. In the coming years, the engagement of pharmacists and microbiologists needs to increase and experts in data management and analysis should also be incorporated into the PROA teams.

Interestingly, only 15 hospitals (58%) included a pharmacist in the pediatric PROA team, and only 14 (54%) a microbiologist. The success of PROA programs lies in their multidisciplinary nature, built around a nucleus comprising an infectious disease specialist or a clinician who is an expert in pediatric infectious diseases, a clinical pharmacist who is an expert in antimicrobial use, and a microbiologist who is an expert in antimicrobial resistance [5,6,7,25]. All team members must be able to invest the time needed to be able to pursue and achieve their goals.

However, the heterogeneity of the pre-established pediatric PROA indicators at the hospitals highlights the need to achieve a consensus with regard to their definition. More attention must also be paid to their monitoring. In this context, the VINCat pediatric PROA has an important role to play in creating a standardized system for recording antimicrobial consumption and antibiotic susceptibility in Catalonia. It promotes the selection and registration of specific pediatric clinical indicators and disseminates the actions and results obtained.

This study has several limitations that should be mentioned. Although the survey evaluated the presence of various indicators and interventions, it did not evaluate all the ones currently in use, or the intensity or frequency of their implementation. In addition, the voluntary nature of the survey administered to the members of the PROA groups means that the results may be overestimated.

In conclusion, the majority of hospitals in Catalonia have implemented the actions recommended in the pediatric antimicrobial stewardship programs, but they have not set up a specific pediatric program of their own. Concrete actions and indicators have yet to be defined, both with regard to the use of antimicrobials and with regard to clinical aspects. These programs should be provided with more resources to allow them to develop their own actions and establish specific indicators for this age group.

The evolution of pediatric antimicrobial stewardship programs in our environment should be monitored over the next few years. Their growth and development will depend on the professionals themselves, the support of institutions and of the VINCat pediatric PROA group, and the resources available.

## 4. Material and Methods

A descriptive observational study was carried out using an electronic questionnaire entitled “The VINCat pediatric PROA SHARP”, prepared from the translation and adaptation of the survey known as “The Sharing Antimicrobial Reports for Pediatric Stewardship (SHARPS)” [20], with the prior consent of the original authors. The multicenter SHARPS survey, conducted in the US in 2016, evaluated the implementation of the seven prerequisites of an antimicrobial stewardship program in the hospital setting [4,20].

The PROA program represented a sustained effort by a health institution to optimize the use of antimicrobials in hospitalized patients, with the aims of improving the clinical results of patients with infections, minimizing the adverse effects associated with the use of antimicrobials (including the emergence and spread of resistance), and guaranteeing the use of cost-effective treatments. The pediatric antimicrobial stewardship team at each hospital comprised pediatricians, microbiologists, and pharmacists with scientific and professional leadership capacity in the use of antimicrobials and in the understanding of bacterial resistance in the field of pediatrics [6]. Depending on the center, other health professionals such as surgeons, intensivists and nurses were also included in the team.

The survey comprised 65 items assessing aspects of the pediatric antimicrobial stewardship programs in operation in hospitals in Catalonia. They were contacted through an email sent to all the members of the VINCat pediatric PROA group. This email contained the web link that hosted the survey. The survey remained open for two months (December 2020 and January 2021).

For subsequent comparisons between centers, the hospitals were classified into three levels of complexity, with a minimum level of pediatric care defined for each one:

Level 1: hospital with a low-complexity general pediatric ward and an emergency department.

Level 2: regional reference hospital with a pediatric ward, an emergency department, non-urgent pediatric surgery, and a neonatal intensive care unit (NICU) for the care of newborns of ≥28 gestational weeks.

Level 3: regional reference hospital offering specialized care (most pediatric specialties), pediatric surgery, and critical care facilities (i.e., NICU and pediatric intensive care units (PICU)).

The results of the survey were analyzed by the core group using the EXCEL computer program (Microsoft^®^ Excel^®^ 2011 version 14.7.7). The descriptive analysis of the variables present in the survey consisted of frequencies and proportions for the qualitative variables, and of medians and interquartile ranges (IQR) for the quantitative variables.

Since the study did not contain personal data, it was unnecessary to obtain the approval of the ethics committee or the informed consent of the members of the participating PROA teams.

## Figures and Tables

**Figure 1 antibiotics-12-00250-f001:**
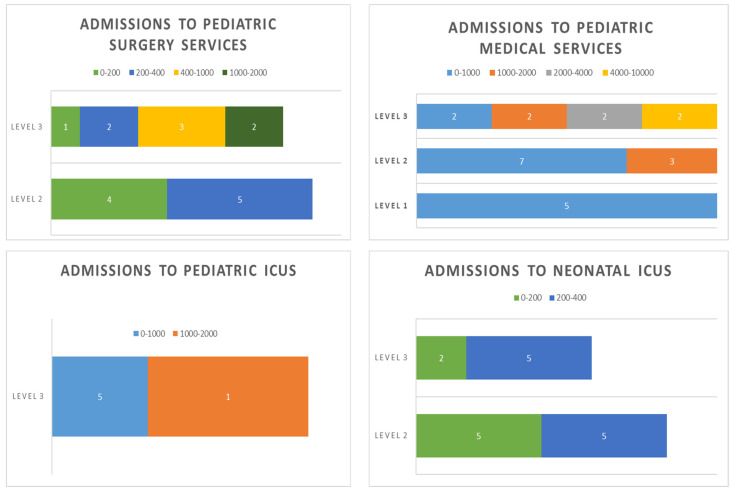
Number of admissions to the hospitals participating in the VINCat pediatric PROA in 2019.

**Figure 2 antibiotics-12-00250-f002:**
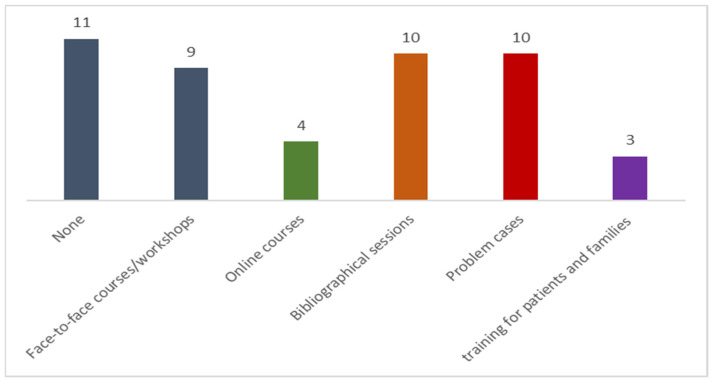
PROA training activities at the hospitals participating in the VINCat pediatric PROA (n = 26).

**Table 1 antibiotics-12-00250-t001:** Structure indicators of the participating hospitals.

	Hospitals Affiliated to the VINCAT-Pediatric PROA (n = 26)
Pediatric service (n; %)	
Presence of a pediatrician in the pediatric PROA	21/26 (81%)
Number of pediatricians with full time dedication to the pediatric PROA ▪ one ▪ two or more	2/26 (8%) 1/26 (4%)
Number of pediatricians specializing in infectious diseases in the pediatric PROA ▪ one ▪ two or more	15/26 (58%) 2/26 (8%)
Pharmacy service (n; %)	
Presence of a pharmacist devoted to the pediatric PROA	15/26 (58%)
Number of pharmacists with full time dedication to the pediatric PROA ▪ one	0/26 (0%)
Microbiology service (n; %)	
Presence of a microbiologist devoted to the pediatric PROA	14/26 (54%)
Number of microbiologists with full time dedication to the pediatric PROA ▪ one	0/26 (0%)
Presence of a microbiology laboratory at the center	24/26 (92%)
Performance of viral PCR at the laboratory at the center	21/26 (81%)
Performance of microbiological diagnostic tests at the laboratory at the center	22/26 (85%)
Electronic reports	26/26 (100%)
Radiology service (n; %)	
Presence of pediatric radiologists	16/26 (62%)
24-h access to CT 7 days a week at the same center	24/26 (92%)
24-h access to MRI 7 days a week at the same center	12/26 (46%)
Electronic reports	25/26 (96%)

**Table 2 antibiotics-12-00250-t002:** Process indicators of pediatric PROA incorporated in the services of the participating hospitals.

	Hospitals Affiliated to the VINCat-Pediatric PROA (n = 26)
Pharmacy service (n; %)	
No process indicators	5/26 (19%)
Monitoring of antimicrobial use	
Clinical audit	6/26 (23%)
Creation of protocols	6/26 (23%)
Microbiology service (n;%)	
No process indicators	7/26 (27%)
Monitoring of antimicrobial susceptibility	16/26 (62%)
Clinical audit	4/26 (15%)
Creation of protocols	5/26 (19%)

**Table 3 antibiotics-12-00250-t003:** Specific PROA actions at the hospitals participating in the VINCat pediatric PROA.

	YES	NO
Possible audits are carried out with feedback for professionals	62%	38%
Restrictive measures are applied in prescriptions	42%	58%
Protocols are reviewed and updated	96%	4%
Presence of a protocol for surgical prophylaxis	54%	46%
Presence of an anti-infection guide for community infections in pediatrics	73%	27%
The prescription of certain antimicrobials must be justified in the computer program	54%	46%
Results of patient’s point of care tests are obtained	58%	42%
Antimicrobials are prescribed with a specific end date	62%	38%
The activities of the pediatric PROA are presented in an annual report or similar document	23%	77%

## Data Availability

Not applicable.

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
