# Peer review of "Assessment of the Plans to Optimize Antimicrobial Use in the Pediatric Population in Catalan Hospitals: The VINCat Pediatric PROA SHARP Survey"

_antibiotics, 2023, doi:10.3390/antibiotics12020250_

Round 1

Reviewer 1 Report

This is a well-written description/summary of the current state of pediatric antimicrobial stewardship in Catalonia. Most relevant references are included, the data are presented in a clear and concise fashion. The discussion has a bit of results review that could potentially be streamlined, but I don't think this is strictly necessary for publication. Tables and figures are helpful and easy to read. 

Author Response

Thank you for your comments. I'm glad to read that the article is attractive to your interest.

Reviewer 2 Report

The current manuscript is not suitable enough to be published. Moreover, it is partly confusing exposition, essentially concerning the results and discussion.  Therefore, the reviewer draws attention to the following issues:

Line 1-4 (title): Plans – what plans? It is not to assess the present status (???) – Why is the word SHARP in the title? It has been briefly addressed in "the Material and Methods" item.

Line 86: Considering Rodrigez-Baño et al., PROA are programs for optimising the use of antibiotics in hospitals. Why do you write this word in the plural? (PROAs)

Line 94-101: has VINCat been integrated into the PROA program?

Line 109-111: The idea is that this program should serve as a starting point for detecting possible areas needing improvement, standardising indicators and defining interventions able to address existing problems. Program: VINCat pedriaticPROA (???)

Line 114-115: Survey or the questionnaire? Centres or hospitals? (the spelling of centers is a non-English variant)

Line 129: The numbers depicted in the bar charts should be explained.

Line 222–231: It should be better discussed. Currently, the results? How did the authors obtain this information?

Line 235–244: It should be better discussed. Please, these statements should be exemplified with the observed results

Line 302-305: This information does not belong to this item

Line 311: Are items questions?

Line 314: Survey or questionnaire? A survey is a method of data collection in which information is gathered through an electronic questionnaire. Is it not?

Line 326: survey data?

Line 327-329: Results from these statements (????)

Reviewer 3 Report

The article titled:

Assessment of the plans to optimize antimicrobial use in the pediatric population in Catalan hospitals: the VINCat pediatric PROA SHARP survey,

by Borja Guarch-Ibáñez et al is the description of status of the implementation f some kind of pediatric antibiotic stewardship in Catalan hospitals, based on the response of SHARPS (Sharing Antimicrobial Reports for Pediatric Stewardship) performance questionnaire containing 65 items. From the answers they established the current status of the implementation of PROA (PRogramas de Optimización de 86 Antimicrobianos).

The introduction of pediatric antibiotic stewardship would be important and utilized in a standardized way in every hospitals where it is implemented.. This activity is clearly describrd in the position paper of Goycochea-Valdivia WA, Melendo Pérez S, Aguilera-Alonso D, Escosa-Garcia L, Martínez Campos L, Baquero-Artigao F; Grupo de Trabajo PROA de la Sociedad Española de Infectología Pediátrica (SEIP). Position statement of the Spanish Society of Paediatric Infectious Diseases on the introduction, implementation and assessment of antimicrobial stewardship programmes in paediatric hospitals. An Pediatr (Engl Ed). 2022 Nov;97(5):351.e1-351.e12. doi: 10.1016/j.anpede.2022.09.007. Epub 2022 Oct 13. PMID: 36243665.

The article of Borja Guarch-Ibáñez et al clearly indicate the very different stewardship activity in the hospitals answering the questionnaire. It does not mention any data that the implementation of PROA has improved anything in the given hospital regarding the utilization of different antibiotic categories (access, watch, reserve) or it had any influence on antibiotic resistance, but probably, it had not been included in the questionnaire.

Some paragraphs (data) dealing with the usefulness of implementing PROA, might improve the scientific importance of the article, instead of this description.

Round 2

Reviewer 2 Report

2. Results

Please start the sentence with the wording "Twenty six" and not 26

Author Response

Dear reviewer, I corrected it on the latest version as you suggested in your comments.   Kind regards

Reviewer 3 Report

The article has been improved, and only focuses attention on the existence or absence of pediatric antibiotic stewardship in the participating hospitals of a survey.  As a description of a situation is appropriate.

Author Response

(The authors gave the same response as above.)
